# Genetic dissection of Down syndrome-associated congenital heart defects using a new mouse mapping panel

Eva Lana-Elola[1], Sheona Watson-Scales[1], Amy Slender[1], Dorota Gibbins[1], Alexandrine Martineau[1], Charlotte Douglas[1], Timothy Mohun[1], Elizabeth MC Fisher[2], Victor LJ Tybulewicz[1,3]*

[1]The Francis Crick Institute, London, United Kingdom; [2]Department of Neurodegenerative Disease, UCL Institute of Neurology, London, United Kingdom; [3]Imperial College London, London, United Kingdom

**Abstract** Down syndrome (DS), caused by trisomy of human chromosome 21 (Hsa21), is the most common cause of congenital heart defects (CHD), yet the genetic and mechanistic causes of these defects remain unknown. To identify dosage-sensitive genes that cause DS phenotypes, including CHD, we used chromosome engineering to generate a mapping panel of 7 mouse strains with partial trisomies of regions of mouse chromosome 16 orthologous to Hsa21. Using high-resolution episcopic microscopy and three-dimensional modeling we show that these strains accurately model DS CHD. Systematic analysis of the 7 strains identified a minimal critical region sufficient to cause CHD when present in 3 copies, and showed that it contained at least two dosage-sensitive loci. Furthermore, two of these new strains model a specific subtype of atrio-ventricular septal defects with exclusive ventricular shunting and demonstrate that, contrary to current hypotheses, these CHD are not due to failure in formation of the dorsal mesenchymal protrusion.

**\*For correspondence:**
Victor.T@crick.ac.uk

**Competing interests:** The author declares that no competing interests exist.

## Introduction

The formation of a functional four-chambered heart is a complex process and perturbations in its development can lead to congenital heart defects (CHD). These affect almost 1% of the population and are a major cause of morbidity and infant mortality (*Fahed et al., 2013*). The chance of being born with CHD is drastically increased (to ~50%) in DS (*Vis et al., 2009*) in which a range of heart defects is seen, including ventricular septal defects (VSD) and outflow tract abnormalities such as overriding aorta (OA) and double outlet right ventricle (DORV). Notably, defects that affect the atrioventricular (AV) junction, especially atrio-ventricular septal defects (AVSD) (*Freeman et al., 1998*) are often seen in DS. AVSD comprise a spectrum of cardiac malformations characterized by a common AV junction, guarded by an essentially common valve, as opposed to separate AV junctions guarded by mitral and tricuspid valves. People diagnosed with AVSD can present with communication between left and right heart chambers (shunting) at the atrial level (ostium primum defect) or at the ventricular level or with shunting at both atrial and ventricular levels (*Mahle et al., 2006*).

The process of forming the AV junction that divides the embryonic heart into a four-chambered structure involves the growth and fusion of a number of tissues from different precursor populations (*Webb et al., 1998*). These include the atrial and ventricular septa (mainly of myocardial origin), the mesenchymal cap of the atrial septum, and the superior and inferior endocardial cushions (endocardial origin). In addition, the dorsal mesenchymal protrusion (DMP) first described as the 'spina vestibuli' by Wilhelm His the elder in 1880 (*His, 1880*; *Mommersteeg et al., 2006*) makes an important contribution to septation of the AV junction (*Anderson et al., 2015*; *Snarr et al., 2007a*;

**eLife digest** Down syndrome is a condition caused by having an extra copy of one of the 46 chromosomes found inside human cells. Specifically, instead of two copies, people with Down syndrome are born with three copies of chromosome 21. This results in many different effects, including learning and memory problems, heart defects and Alzheimer's disease. Each of these different effects is caused by having a third copy of one or more of the approximately 230 genes found on chromosome 21. However, it is not known which of these genes cause any of these effects, and how an extra copy of the genes results in such changes.

Now, Lana-Elola et al. have investigated which genes on chromosome 21 cause the heart defects seen in Down syndrome, and how those heart defects come about. This involved engineering a new strain of mouse that has an extra copy of 148 mouse genes that are very similar to 148 genes found on chromosome 21 in humans. Like people with Down syndrome, this mouse strain developed heart defects when it was an embryo.

Using a series of six further mouse strains, Lana-Elola et al. then narrowed down the potential genes that, when in three copies, are needed to cause the heart defects, to a list of just 39 genes. Further experiments then showed that at least two genes within these 39 genes were required in three copies to cause the heart defects.

The next step will be to identify the specific genes that actually cause the heart defects, and then work out how a third copy of these genes causes the developmental problems.

*Snarr et al., 2007b*; *Webb et al., 1998*). The DMP is a mesenchymal structure at the venous pole of the developing heart derived from the second heart field (SHF) and perturbations of its development result in AVSD (*Briggs et al., 2013*; *Goddeeris et al., 2008*; *Rana et al., 2014*; *Tian et al., 2010*; *Webb et al., 1999*; *Xie et al., 2012*). Abnormalities in the DMP have also been detected in human fetal hearts with trisomy 21, leading to the hypothesis that malformation of this tissue causes the AVSD in DS (*Blom et al., 2003*).

Hsa21 carries 233 protein-coding genes (genome assembly GRCh38.p5) and it is thought that DS phenotypes result from an increased dosage of one or more of the genes on Hsa21. The search for dosage-sensitive genes that when present in 3 copies cause DS phenotypes has been approached using both human and mouse genetics. In humans, rare cases of partial trisomy 21 have been used to identify critical regions that contain dosage-sensitive genes that when present in 3 copies cause DS phenotypes (*Delabar et al., 1993*; *Korbel et al., 2009*; *Korenberg et al., 1994*; *Lyle et al., 2009*). Alternatively, mouse strains have been generated to study the pathology and genetics of DS.

Hsa21 shares synteny with a large region on mouse chromosome 16 (Mmu16) and with shorter regions on Mmu10 and Mmu17. The first two DS models generated, Ts65Dn (*Davisson et al., 1990*) and Ts1Cje (*Sago et al., 1998*) have duplications of regions of mouse chromosome 16 (Mmu16) that are orthologous to Hsa21, and have been used to identify dosage-sensitive genes contributing to some DS phenotypes (*Baek et al., 2009*; *Chakrabarti et al., 2010*; *Lana-Elola et al., 2011*; *Salehi et al., 2006*; *Sussan et al., 2008*). However, both models also have additional aneuploidy (trisomy of 60 genes on Mmu17 in Ts65Dn and monosomy of 7 genes on Mmu12 in Ts1Cje mice), making interpretation of their phenotypes difficult (*Duchon et al., 2011*). Another DS model, the Tc1 mouse, carries a freely-segregating copy of Hsa21 and shows many DS phenotypes, including CHD (*Dunlevy et al., 2010*; *O'Doherty et al., 2005*), although the Hsa21 in this strain is not intact and the mice are mosaic for the human chromosome, again making interpretation difficult (*Gribble et al., 2013*). Thanks to recent advances in chromosome engineering, a number of mouse strains with duplications of regions of mouse chromosomes orthologous to Hsa21 have been generated, resulting in partial trisomies, including the most complete mouse model for DS to date (Dp(16) 1Yey/+;Dp(17)1Yey/+;Dp(10)1Yey/+), which carries a duplication of all Hsa21-orthologous regions on Mmu16, Mmu17 and Mmu10 (*Brault et al., 2015*; *Li et al., 2007*; *Liu et al., 2013*; *Liu et al., 2011*; *Olson et al., 2004*; *Pereira et al., 2009*; *Yu et al., 2010*). However, a comprehensive mapping panel to finely map dosage-sensitive genes in DS has not been available until now.

To identify dosage-sensitive critical regions and candidate genes, we now report the generation of a fine mapping panel of 7 partial trisomies of Mmu16 that can be used to identify the genetic

basis of DS phenotypes where such genes reside in the region of Mmu16 orthologous to Hsa21. We use this panel to investigate the cardiac defects in DS, employing high-resolution episcopic microscopy (HREM) to analyze the precise three-dimensional (3D) morphology of the developing hearts. This combined approach enabled us to narrow down the critical region for CHD in DS to a minimal region containing 39 protein-coding genes and to show that this region contains at least two dosage-sensitive loci required in three copies to cause CHD. Furthermore using dual-wavelength HREM we show that the DMP develops normally in these DS models, and conclude that CHD in these DS models are not caused by failure in formation or growth of the DMP.

## Results

### A mouse genetic mapping panel for DS

To expedite the identification of dosage-sensitive genes required in 3 copies to cause DS phenotypes, we generated a novel high-resolution 'mapping panel' of 7 strains with duplications in Mmu16 (*Figure 1a*). Of the protein-coding genes on Hsa21 that have orthologues in the mouse, the largest fraction (~58%) is located in the telomeric region of Mmu16. Thus we used long-range Cre/loxP mediated recombination to engineer the Dp1Tyb mouse strain carrying a duplication from *Lipi* to *Zbtb21* on Mmu16 spanning 23 Mb and 148 coding genes. We then generated 3 further strains with contiguous segmental duplications completely covering the region duplicated in Dp1Tyb: Dp9Tyb (from *Lipi* to *Hunk*), Dp2Tyb (from *Mis18a* to *Runx1*) and Dp3Tyb (from *Mir802* to *Zbtb21*). To increase the resolution of the mapping panel further, we generated another 3 strains with duplications breaking up the telomeric region of Mmu16 into three contiguous fragments completely covering the region duplicated in the Dp3Tyb strain: Dp4Tyb (from *Mir802* to *Dscr3*), Dp5Tyb (from *Dyrk1a* to *B3galt5*) and Dp6Tyb (from *Igsf5* to *Zbtb21*).

We recovered live mutant mice from all 7 strains but noted that the yield of Dp1Tyb and Dp3Tyb mice was significantly reduced by 50% and 25% respectively (*Table 1*). We also observed hydrocephalus in Dp1Tyb mice around the time of weaning, but not in any other strain (not shown). Comparative genome hybridization (CGH) confirmed the expected copy number increase (from 2 to 3) across the duplicated regions of Mmu16 in all 7 strains, with no other copy number changes seen in the genome (*Figure 1b* and data not shown). These new strains provide a unique resource to study DS-associated phenotypes and to map dosage-sensitive genes causing these phenotypes.

### Dp1Tyb mice show cardiac defects similar to those seen in DS

To investigate whether duplication of the region of Mmu16 from *Lipi* to *Zbtb21* in the Dp1Tyb strain was sufficient to cause CHD, we used HREM and 3D modeling (*Weninger et al., 2006*), an approach we had previously used successfully to identify CHD in the Tc1 strain (*Dunlevy et al., 2010*). These methods are particularly suited to examination of complex 3D structures, such as the developing heart, overcoming limitations of conventional 2D histological methods. We observed a significant increase of CHD in E14.5 Dp1Tyb embryos compared to their wild-type (Wt) littermates (*Figure 2a*). Detailed examination of heart morphology in the mutant embryos revealed a range of defects (*Table 2*). About 18% of Dp1Tyb hearts show abnormal arterial trunk arrangements such as OA or DORV with subaortic communication and 62% had a VSD either alone or in combination with other defects (*Figure 2b,c* and *Videos 1* and *2*). Two subtypes of VSD were observed: perimembranous VSD (pVSD), located in the membranous portion of the ventricular septum and muscular or trabecular VSD (mVSD), which opens to the inlet of the right ventricle (*Figure 2c* and *Videos 3* and *4*). Around 25% of Dp1Tyb embryos displayed AVSD presenting two bridging leaflets across the single AV junction and an 'unwedged' morphology of the left outflow tract (*Figure 2c* and *Video 5*). Notably, the AVSD in Dp1Tyb mice were associated exclusively with a ventricular shunt and never with an atrial shunt (*Video 6*). Thus Dp1Tyb model a subtype of AVSD with a ventricular component, in which the cushions are attached to the leading edge of the atrial septum. Overall, these data show that the Dp1Tyb mouse models the main types of CHD seen in DS.

### CHD mapped to at least 2 loci in a 4.9 Mb telomeric region of Mmu16

To identify the critical region(s) sufficient to cause CHD in DS, we examined heart development in Dp9Tyb, Dp2Tyb and Dp3Tyb mice that between them have duplications covering the entire

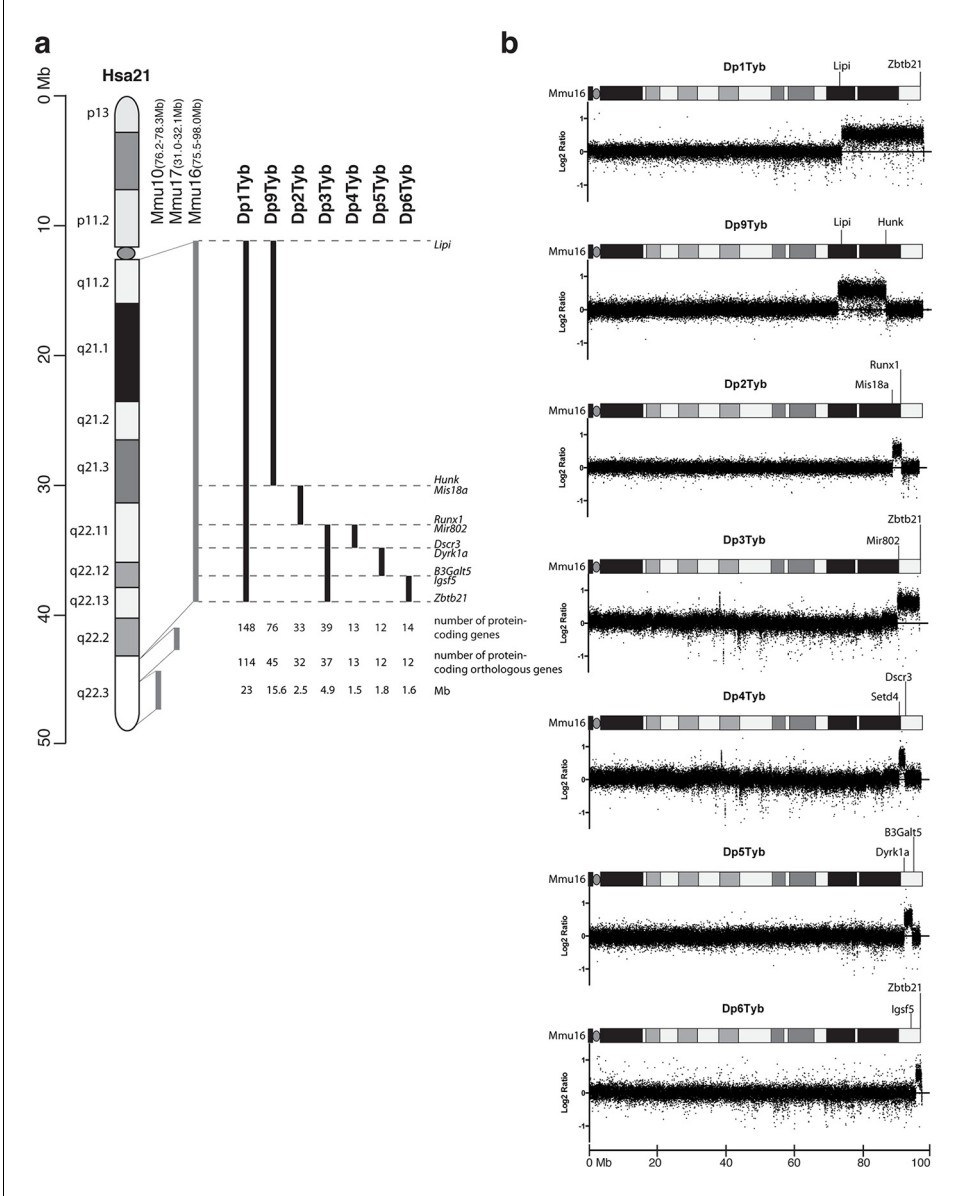

**Figure 1.** Generation of a mouse mapping panel of partial trisomies for DS phenotypes. (a) Representation of Hsa21 and the regions of conserved synteny with orthologous regions on Mmu10, Mmu17 and Mmu16 (grey lines), indicating the coordinates of each mouse region. Diagram of Hsa21 shows the main cytogenetic bands (rectangles of different colors) and the centromere (oval). Black lines show the extent of the duplication in each of the mouse strains reported here, indicating the first and last genes at the ends of each duplication. The number of genes and extent in Megabases (Mb) of each duplicated region was calculated as described in Methods (b) CGH analysis of each duplication strain (2 mice analyzed/strain). Plots show log2-transformed hybridization ratios for DNA from each duplication strain versus C57BL/6JNimr. The duplicated regions would be expected to have a 1.5-fold increase in DNA content (log2 ratio = 0.58).

duplicated region in Dp1Tyb mice (*Figure 1a*). HREM analysis of embryonic hearts at E14.5 revealed that neither Dp9Tyb nor Dp2Tyb showed a significant increase of CHD compared to Wt littermates (*Figure 3a*). In contrast, the telomeric duplication of 39 coding genes in the Dp3Tyb strain was sufficient to cause a significant increase in embryos with CHD. We observed VSD such as pVSD, mVSD in the inlet portion of the ventricular septum, as well as AVSD (*Figure 3b*, *Table 2* and *Videos 7* and *8*). We noted a reduced frequency of outflow tract defects in Dp3Tyb embryos compared to Dp1Tyb, however this difference was not statistically significant. Overall the types of defects

**Table 1.** Yields of wild-type (WT) and mutant mice from the 7 duplication strains.
Table shows the numbers and percentages of wild-type and mutant mice recovered at weaning (~3 weeks old) from each of the 7 duplication strains which were bred by crossing a mutant mouse with a C57BL/6JNimr mouse. The numbers were tested for significant difference from the expected Mendelian yields (wild-type:mutant, 50%:50%) using a 2-tailed Fisher's exact test and p-values are reported in the final column where these were <0.05, otherwise are indicated as not significant (ns).

| | Mouse numbers | | Percentages | | |
| --- | --- | --- | --- | --- | --- |
| Strain | WT | Mutant | WT | Mutant | p-value |
| Dp1Tyb | 427 | 208 | 67.24% | 32.76% | <0.0001 |
| Dp2Tyb | 305 | 253 | 54.66% | 45.34% | ns |
| Dp3Tyb | 223 | 165 | 57.47% | 42.53% | 0.0437 |
| Dp4Tyb | 101 | 78 | 56.42% | 43.58% | ns |
| Dp5Tyb | 99 | 74 | 57.23% | 42.77% | ns |
| Dp6Tyb | 191 | 157 | 54.89% | 45.11% | ns |
| Dp9Tyb | 97 | 91 | 51.60% | 48.40% | ns |

observed in Dp3Tyb were very similar in both type and severity to the ones in Dp1Tyb mice, suggesting that all the dosage-sensitive gene(s) required to cause CHD in Dp1Tyb mice were located in this shorter region.

To further narrow down the critical region for CHD, we analyzed embryonic hearts from the Ts1Rhr strain, which has a duplication that is shorter than that in the Dp3Tyb strain by just 8 genes (7 coding genes and 1 micro-RNA gene) (*Olson et al., 2004*). We and others have previously reported that this strain does not show CHD (*Dunlevy et al., 2010*; *Liu et al., 2011*). However, in our earlier study we had examined Ts1Rhr on the mixed 129S8;C57BL/6JNimr genetic background, whilst Dp1Tyb and Dp3Tyb are maintained on the C57BL/6JNimr background. Thus in order to eliminate possible confounding effects of background differences, we backcrossed Ts1Rhr to the C57BL/6JNimr background and used HREM to look for CHD. Once again we found no increased frequency of CHD in the Ts1Rhr strain (*Figure 3c*). Thus an extra copy of one or more of the 8 genes duplicated in Dp3Tyb but not in Ts1Rhr is required to cause CHD. Next we analyzed Dp4Tyb, Dp5Tyb and Dp6Tyb mice each of which carry a duplication that between them cover the whole of the duplicated region in Dp3Tyb mice. Remarkably, none of these 3 strains showed significant increased rates of CHDs indicating that there are at least two dosage-sensitive loci in the Dp3Tyb mouse that contribute to CHD (*Figure 3c*).

Taken together these data show that a 4.9 Mb region of Mmu16 from *Mir802* to *Zbtb21* is sufficient when in 3 copies to generate cardiac defects similar to those seen in DS. Furthermore, the mapping analysis shows that there are two or more loci within this region that are required in 3 copies to cause CHD, and that at least one of these resides within the 8 genes duplicated in Dp1Tyb but not Ts1Rhr mice.

## Development of the DMP

The DMP plays a crucial role in the formation of the AV junction, and defects in its development have been proposed to underlie the AVSD in DS (*Blom et al., 2003*; *Briggs et al., 2012*). In order to assess if DMP development was perturbed in the Dp1Tyb mouse model of DS, we first established a method to follow its development during formation of the AV junction. The *Isl1* gene is expressed in the SHF and also in the DMP which is derived from it and hence can be used as a marker for this tissue. We visualized expression of *Isl1* using 2 different mouse strains: the Isl1Cre (*Cai et al., 2008*) strain crossed to Rosa26RLacZ reporter mice (*Soriano, 1999*) to identify cells that are expressing or had expressed *Isl1*, and thus are derived from the SHF; and the Isl1nLacZ strain (*Sun et al., 2007*) to visualize ongoing expression of *Isl1*. Gene expression studies have traditionally relied on staining individual sections and rendering a 3D expression pattern by compiling histological sections, but this approach results in loss of resolution and is constrained by the chosen sectioning plane. To get around this, we utilized dual-wavelength HREM (*Mohun and Weninger, 2011*) to visualize β-

**Table 2.** Cardiovascular abnormalities in E14.5 embryos.
Table shows the numbers of different cardiac defects found in embryos from the indicated duplication strains and in Wt littermate controls.

| Types of defects | | | Dp1Tyb | | Dp9Tyb | | Dp2Tyb | | Dp3Tyb | | Ts1Rhr | | Dp4Tyb | | Dp5Tyb | | Dp6Tyb | |
|---|---|---|---|---|---|---|---|---|---|---|---|---|---|---|---|---|---|---|
| | | | Wt | Dp1Tyb | Wt | Dp9Tyb | Wt | Dp2Tyb | Wt | Dp3Tyb | Wt | Ts1Rhr | Wt | Dp4Tyb | Wt | Dp5Tyb | Wt | Dp6Tyb |
| Single defects | Septal defects | ASD | 1 | 0 | 0 | 0 | 0 | 0 | 1 | 0 | 0 | 0 | 0 | 0 | 0 | 0 | 0 | 0 |
| | | pVSD | 3 | 9 | 0 | 1 | 2 | 5 | 2 | 0 | 4 | 1 | 0 | 4 | 1 | 1 | 4 | 1 |
| | | mVSD | 0 | 0 | 1 | 0 | 0 | 2 | 0 | 4 | 0 | 0 | 0 | 0 | 0 | 0 | 2 | 1 |
| | OFT defects | OA | 0 | 0 | 0 | 0 | 0 | 0 | 0 | 0 | 0 | 0 | 0 | 0 | 0 | 0 | 0 | 0 |
| | | DORV | 0 | 0 | 0 | 0 | 0 | 0 | 0 | 0 | 0 | 0 | 0 | 0 | 0 | 0 | 0 | 0 |
| | AVSD | | 0 | 1 | 0 | 0 | 0 | 0 | 0 | 0 | 0 | 0 | 0 | 0 | 0 | 0 | 0 | 0 |
| Multiple defects | OFT + VSD | | 1 | 2 | 0 | 0 | 0 | 1 | 0 | 0 | 0 | 0 | 0 | 0 | 2 | 0 | 0 | 0 |
| | pVSD + mVSD | | 1 | 3 | 0 | 0 | 0 | 1 | 0 | 0 | 0 | 2 | 0 | 0 | 0 | 0 | 0 | 0 |
| | VSD + AVSD | | 1 | 4 | 0 | 0 | 0 | 0 | 0 | 6 | 1 | 2 | 0 | 0 | 0 | 1 | 0 | 0 |
| | OFT + VSD + AVSD | | 0 | 5 | 0 | 0 | 0 | 0 | 0 | 1 | 0 | 0 | 0 | 0 | 0 | 1 | 0 | 0 |
| Total number of CHD | | | 7 | 24 | 1 | 1 | 2 | 9 | 3 | 11 | 5 | 5 | 0 | 4 | 3 | 3 | 6 | 2 |
| Embryos analyzed | | | 26 | 39 | 22 | 17 | 16 | 26 | 26 | 25 | 20 | 21 | 15 | 19 | 11 | 20 | 28 | 19 |
| % of CHD | | | 26.9 | 61.5 | 4.5 | 5.9 | 12.5 | 34.6 | 11.5 | 44.0 | 25.0 | 23.8 | 0 | 21.0 | 27.2 | 15.0 | 21.4 | 10.5 |

galactosidase expression in the 3D context of the developing heart in order to get an accurate view of the development of the DMP. At E11.5 analysis of both strains shows there is an expansion of *Isl1*-expressing mesenchymal tissue from within the right pulmonary ridge, which forms the DMP and has started to protrude ventrally into the cavity of the right atrium (*Figure 4*). One day later at E12.5, the DMP has protruded more ventrally and it is now in contact with the inferior atrioventricular cushion. At these stages both reporters give a similar picture, as the DMP is actively expressing *Isl1*. Later in development (E13.5) the expression of *Isl1* diminishes and totally disappears by E14.5 (*Figure 4*; *Snarr et al., 2007b*). However development of the DMP can still be followed using the Isl1Cre/Rosa26RLacZ fate reporter strain. This revealed that by E14.5 the DMP forms the ventro-caudal buttress at the core of the AV junction sandwiched between the atrial septum and the endocardial cushions that have now developed into the tricuspid and mitral valves (*Figure 4*). Overall, using two different genetic lineage markers of the SHF, these data show a detailed 3D view of the spatiotemporal development of the DMP.

## The DMP is present and largely unaffected in the Dp1Tyb mouse model for DS

To test whether the AVSD seen in Dp1Tyb mice are caused by defects in development of the DMP, as has been proposed, we imaged the DMP in these mice using the Isl1nLacZ reporter strain. We found that at both E11.5 and E12.5 the DMP was present in Dp1Tyb embryos at a similar location to that seen in Wt mice (*Figure 5a*). Volumetric analysis showed that at E11.5 in both Wt and Dp1Tyb embryos, the DMP appears rounded in shape and attached to the dorsal extracardiac mesenchyme. Furthermore, the size and shape factor of the DMP in Dp1Tyb mice was similar to that in Wt mice. At E12.5 the size of the DMP was reduced in the mutant mice, but in both strains the DMP was more elongated and showed a similar shape factor (*Figure 5a*).

At E11.5 and E12.5 it is not possible to identify which of the hearts would have developed AVSD, and thus we could not tell if mice with largely normal DMP development would have gone on to show defects. To evaluate this issue directly we examined E14.5 Dp1Tyb hearts with AVSD for the presence of the DMP. Once again we used HREM with 3D modeling and examined the same Wt and Dp1Tyb hearts using three different views. In a short axis view across the AV junction we saw an AVSD in the Dp1Tyb heart with the superior and inferior bridging leaflets across the single AV junction while the Wt heart showed a normal AV junction (*Figure 5b*, left panels). A long axis four-

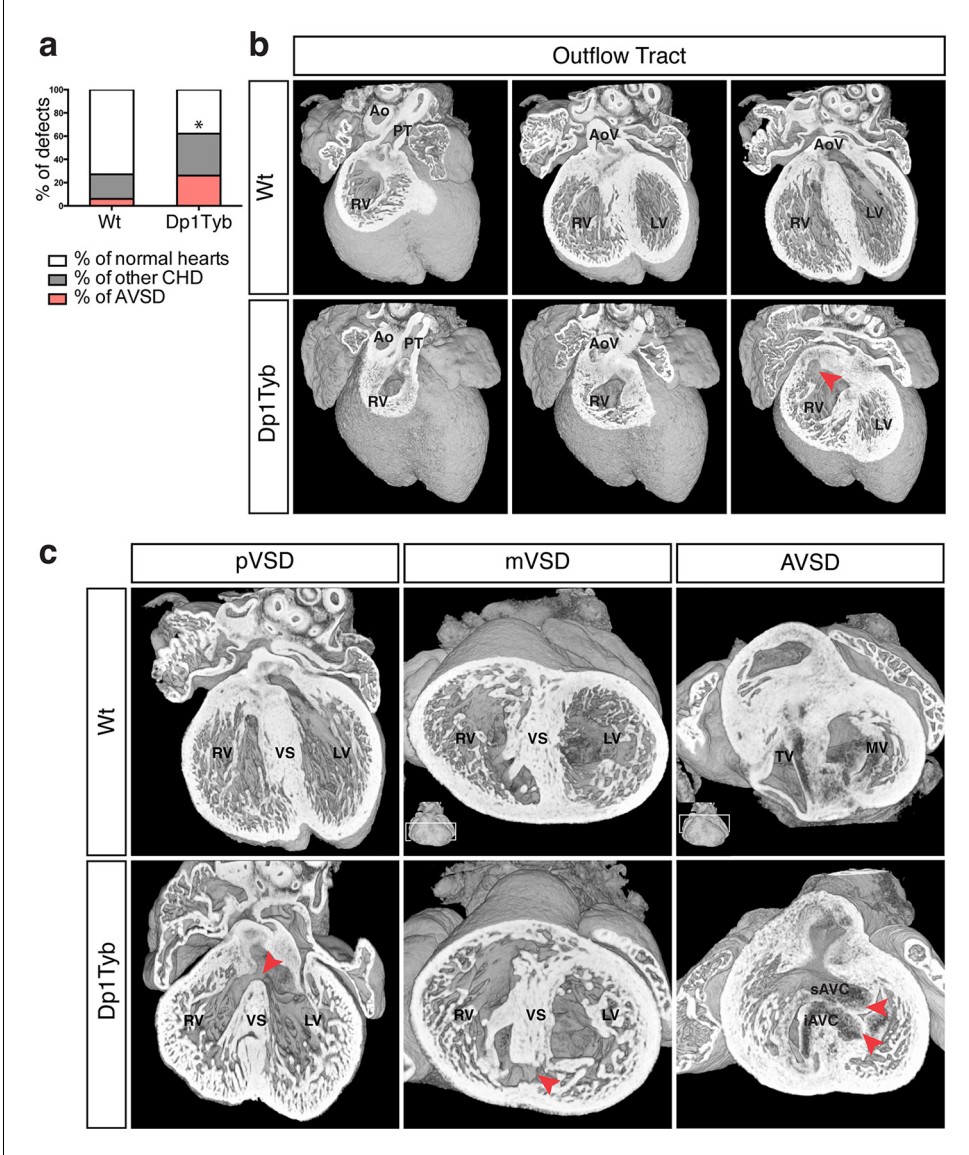

**Figure 2.** Dp1Tyb mice recapitulate the CHD seen in DS. (a) Graph shows the percentage of CHD in Dp1Tyb embryonic hearts at E14.5 compared to wild-type (Wt) littermates; n = 26 Wt and 39 Dp1Tyb embryos. Significant difference to corresponding wild-type incidence (Fisher's exact test) *p<0.05. (b) 3D HREM rendering of Wt and Dp1Tyb hearts, eroded from anterior to posterior to show the four-chamber view in 3 successive planes. Outflow tract defects such as double outlet right ventricle are observed in Dp1Tyb hearts (red arrow head indicates the aortic valve connecting to the right ventricle). (c) 3D reconstructions of Wt and Dp1Tyb E14.5 hearts with different types of CHD (red arrowheads): perimembranous ventricular septal defects (pVSD), muscular ventricular septal defect (mVSD), atrio-ventricular septal defect (AVSD). Ao, aorta; AoV, aortic valve; iAVC, inferior atrio-ventricular cushion; LV, left ventricle; MV, mitral valve; PT, pulmonary trunk; RV, right ventricle; sAVC, superior atrio-ventricular cushion; TV, tricuspid valve; VS, ventricular septum.

chamber view of the same hearts showed an intact ventricular septum in the Wt heart but a VSD in the Dp1Tyb heart (*Figure 5b*, middle panels). Finally, a more dorsal plane of the long axis view of the same hearts showed that the DMP was located in the correct position at the AV junction in both the Wt and the Dp1Tyb hearts (*Figure 5b*, right panels, and *Video 6*). We performed the same analysis on the Dp3Tyb mice and once again found that the DMP was present in embryos with AVSD (not shown). Taken together these data show that the AVSD found in these new mouse models of DS are not caused by perturbations of the growth of the DMP. Importantly, we note that defects in the DMP have been previously shown to result in atrial septal defects or AVSD with atrial shunting

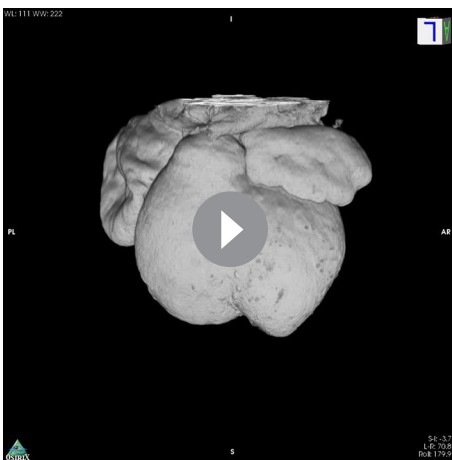

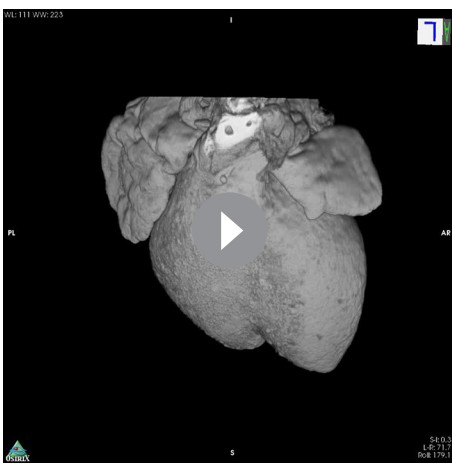

**Video 1.** OA in an E14.5 Dp1Tyb heart. At the start of the video an image of the whole heart is seen and as it zooms in the frontal erosion of the heart reveals the four-chamber view. The pulmonary valves are seen at 00:05, coming from the right ventricle and at 00:08 the aorta is seen positioned directly over a VSD.
https://elifesciences.org/articles/11614#video1

**Video 2.** DORV in an E14.5 Dp1Tyb heart. The video starts with an image of the whole heart and frontal erosion reveals that the aorta is communicating with the right ventricle (00:04). The video then zooms in to reveal in more detail how the aortic valves are communicate with the right ventricle. Further erosion also shows a VSD at the end of the video (00:17).
https://elifesciences.org/articles/11614#video2

(*Briggs et al., 2012*), neither of which are seen in Dp1Tyb or Dp3Tyb mice. In contrast these strains show AVSD with exclusive ventricular shunting, implying that this subtype of AVSD must be due to defects in tissues other than the DMP.

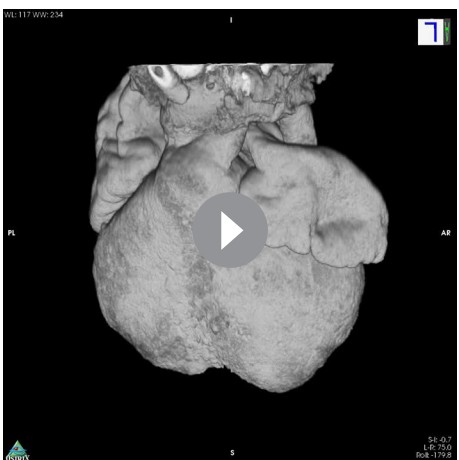

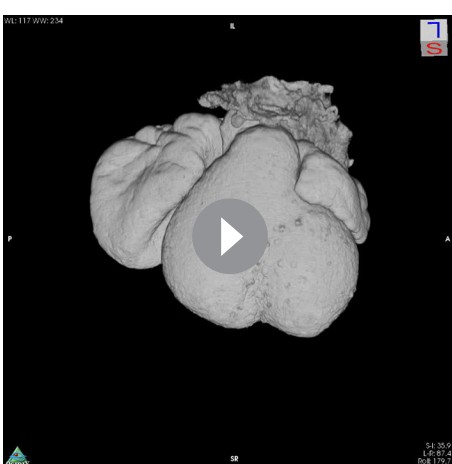

**Video 3.** pVSD in an E14.5 Dp1Tyb heart. At the start the video shows the whole heart and then erodes frontally to show a pVSD (00:08). The video then erodes slightly further to show the connection of the ventricular septum with the endocardial cushions. The image then zooms in and erodes backwards to show the pVSD at higher magnification.
https://elifesciences.org/articles/11614#video3

**Video 4.** mVSD in an E14.5 Dp1Tyb heart. The video shows a whole heart that rotates to view the apex of the ventricles and then erodes transversally revealing the ventricular septum. The mVSD is first seen at 00:08 and the image then zooms in to show it at higher magnification.
https://elifesciences.org/articles/11614#video4

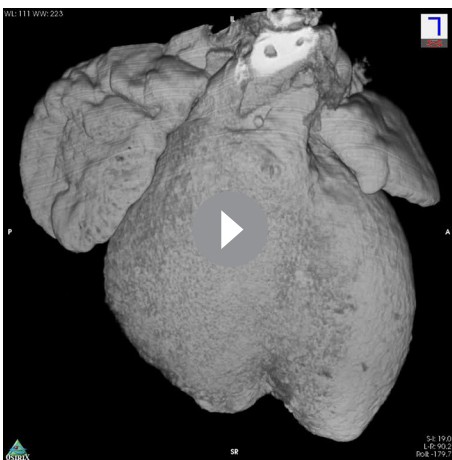

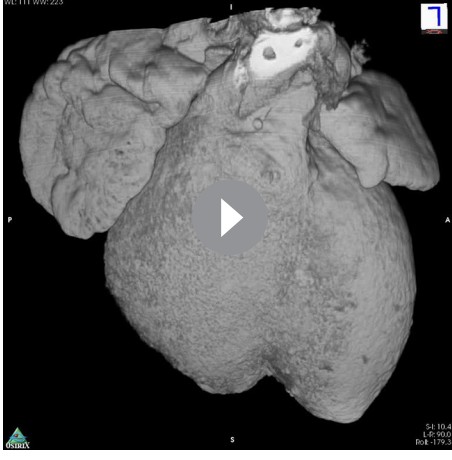

**Video 5.** AVSD in an E14.5 Dp1Tyb heart. The video shows a whole heart that rotates to view the apex of the ventricles and then erodes upwards. The bridging leaflets across the common AV junction can first be seen at 00:05, but they are more clearly visible as the heart is eroded further and a close look at the AV valves is shown at the end of the video (00:16), where the superior and inferior bridging leaflets forms a common AV junction. Note the "unwedged" morphology of the left outflow tract on top of the superior bridging leaflet.
https://elifesciences.org/articles/11614#video5

**Video 6.** AVSD with intact DMP in an E14.5 Dp1Tyb heart. The video shows the whole heart that initially erodes transversally from the apex to reveal an AVSD (00:03 to 00:07). The heart is then reconstructed and eroded frontally to show a pVSD (00:13). Further erosion shows an intact DMP on top of the endocardial cushions (00:13 to 00:17). Note this is the same heart as shown in *Video 5*.
https://elifesciences.org/articles/11614#video6

## Discussion

Cardiac abnormalities are very common in DS. Approximately half of all babies born with DS have a heart defect, many of which are serious and need to be surgically repaired. In order to understand the genetic and molecular mechanisms that lead to CHD in DS it is essential to establish mouse models for DS that accurately recapitulate them. Here we report the generation of a new mouse strain Dp1Tyb with a 23 Mb duplication of the entire region of Mmu16 orthologous to Hsa21, and show that it recapitulates the main types of CHD seen in DS. The Dp(16)1Yey mouse strain carries a similar duplication to Dp1Tyb encompassing the same set of genes and was also reported to have heart defects during embryogenesis (*Li et al., 2007*). Interestingly, the cardiac phenotypes in Dp1Tyb are very similar to the ones we previously observed in the transchromosomic strain Tc1 (*Dunlevy et al., 2010*; *O'Doherty et al., 2005*), despite differences in the Hsa21 orthologous gene content in the two strains and the different species origin for the duplicated genes: mouse in Dp1Tyb mice, human in Tc1 mice. This phenotypic similarity supports the conclusion that the CHD seen in these strains are caused by the same pathological mechanisms as those in people with DS. As with CHD in humans with DS we observed incomplete penetrance of the phenotype in Dp1Tyb mice. In humans this has been ascribed to genetic modifiers (*Li et al., 2012*). However the Dp1Tyb mouse strain was analyzed on an inbred background, suggesting that incomplete penetrance may be caused by stochastic effects during embryonic cardiogenesis. While the recovery of E14.5 Dp1Tyb embryos was consistent with expected Mendelian numbers (not shown), the recovery of Dp1Tyb mice at weaning was reduced to 50% of the expected numbers (*Table 1*). Some of this loss could be due to the most severe CHDs, since AVSDs were found in ~25% of E14.5 Dp1Tyb embryos, but it is likely that there are other unknown causes for this perinatal lethality. The other strain to show AVSDs, Dp3Tyb, showed a 25% reduction in recovery of mice at weaning, similar to the observed frequency of AVSDs at E14.5, making this a likely cause of the reduced yield of mutant mice.

Together with the Dp1Tyb strain we generated a collection of 7 new mouse strains with segmental duplications ranging from 1.5 Mb to 23 Mb that provide a genetic resource for mapping dosage-sensitive genes required in 3 copies to cause DS phenotypes. We used this mapping panel to

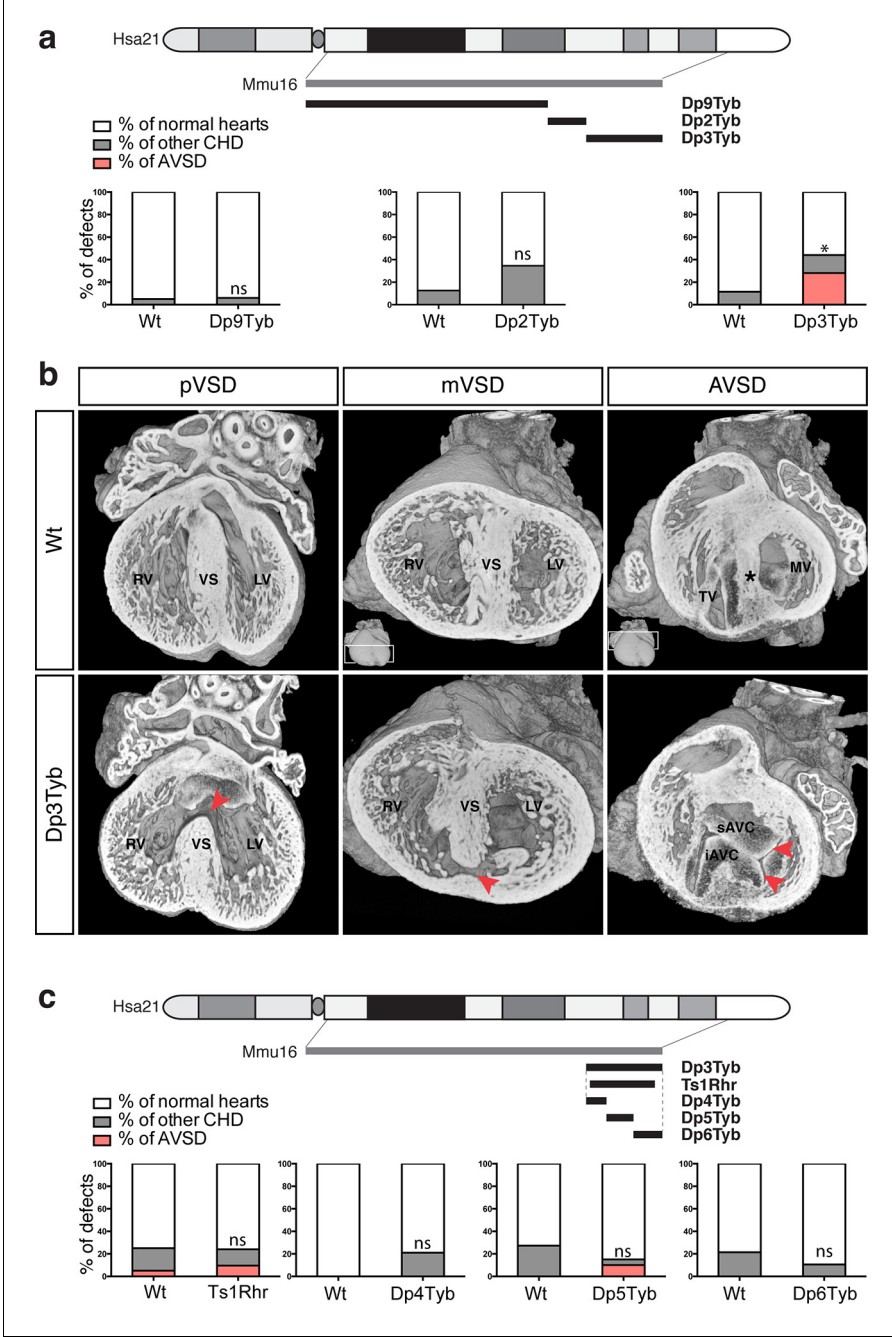

**Figure 3.** Genetic dissection of DS-associated CHD. (a) Representation of Hsa21 showing region of conserved synteny with Mmu16 in grey and the extent of the duplications in Dp9Tyb, Dp2Tyb and Dp3Tyb in black. Graphs show the incidence of CHD in each strain; n = 22 Wt, 17 Dp9Tyb; n = 16 Wt, 26 Dp2Tyb and n = 26 Wt, 25 Dp3Tyb embryos. Significant difference to corresponding wild-type incidence (Fisher's exact test) *$P$<0.05. ns, not significant. (b) 3D reconstructions of Wt and Dp3Tyb E14.5 hearts with different types of CHD (red arrowheads) as described in *Figure 2c*. (c) Representation of Hsa21 showing region of conserved synteny with Mmu16 in grey and the extent of the duplications in the indicated strains in black. Graphs shows incidence of CHD. n = 20 Wt, 21 Ts1Rhr; n = 15 Wt, 19 Dp4Tyb; n = 11 Wt, 20 Dp5Tyb and n = 28 Wt, 19 Dp6Tyb embryos analyzed. ns, not significant (Fisher's exact test). iAVC, inferior atrio-ventricular cushion; LV, left ventricle; MV, mitral valve; RV, right ventricle; sAVC, superior atrio-ventricular cushion; TV, tricuspid valve; VS, ventricular septum.

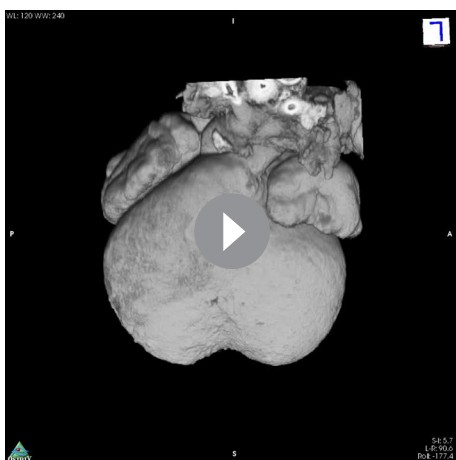

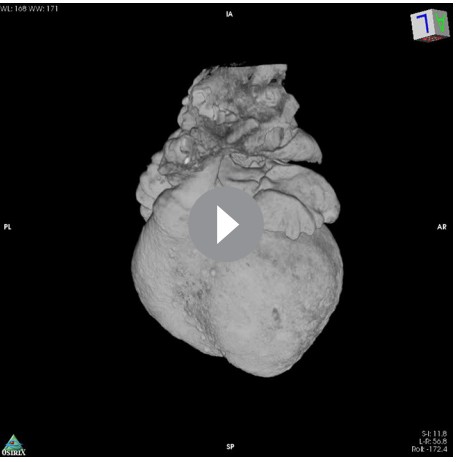

**Video 7.** mVSD in an E14.5 Dp3Tyb heart. The embryonic heart is eroded upwards from the apex of the ventricles to reveal an mVSD in the inlet portion of the ventricular septum (00:17).
https://elifesciences.org/articles/11614#video7

**Video 8.** AVSD in an E14.5 Dp3Tyb heart. The heart is first eroded frontally to show a pVSD (00:04). The heart is then reconstructed and eroded from the top of the OFT downwards to reveal an AVSD (00:13 to 00:17).
https://elifesciences.org/articles/11614#video8

identify a 4.9 Mb genomic region (from *Mir802* to *Zbtb21*) that when present in 3 copies is sufficient to cause CHD. Furthermore, from analysis of shorter duplications, we show that the phenotype is caused by at least two distinct dosage-sensitive loci. By combining this data with analysis of the Ts1Rhr strain, we have determined that, minimally, one of the two loci lies either in an interval at the centromeric end of the duplication in Dp3Tyb (from *Mir802* to *Setd4*) or at the telomeric end (from *Mx2* to *Zbtb21*) (*Figure 6a*). These two segments between them contain only 7 coding genes (*Setd4, Mx2, Tmprss2, Ripk4, Prdm15, C2cd2* and *Zbtb21*) and one microRNA gene (*Mir802*), none of which had been previously implicated in causing CHD in DS. A previous study proposed that a 3.7 Mb genomic region of Mmu16 (*Ifnar1 – Kcnj6*) in Dp(16)4Yey was sufficient to cause CHD (*Liu et al., 2013*); this region overlaps the centromeric end of the duplicated region of Dp3Tyb by 1.9 Mb from *Mir802* to *Kcnj6*, which may help to further narrow down the search for causative genes (*Figure 6b*). We note that while the Dp5Tyb and Ts1Rhr strains did not show a statistically significant increase in CHDs compared to their wild-type littermate controls, in both strains we detected 2 mutant embryos with AVSDs (out of 20 or 21 embryos analyzed respectively). Thus it is possible that these strains have a weak phenotype – a larger number of embryos from these strains would need to be analyzed to establish whether this is significant. If true, it would suggest that one or more of the causative genes may reside within the Dp5Tyb interval (which is entirely contained within the Ts1Rhr interval).

In contrast, analysis of humans with partial trisomies of Hsa21 identified distinct regions that contribute to cardiac abnormalities (*Barlow et al., 2001*; *Korbel et al., 2009*). An individual (PM) trisomic for the 7.7 Mb region from *PSMG1* to *PRMT2* had a VSD, whereas the shortest partial trisomy (individual BA) that gave rise to AVSD extended 10.0Mb from *HLCS* to *PRMT2* (*Figure 6b*). We note that that while both of these intervals overlap substantially with the minimal Dp3Tyb region identified in our studies, the duplicated region in individual PM does not overlap at all with the *Ifnar1 – Kcnj6* region identified by Liu et al as being sufficient to cause CHDs (*Liu et al., 2013*). In particular the human studies suggested that increased dosage of the *DSCAM* gene may be responsible for the CHDs (*Korbel et al., 2009*). In contrast the mouse studies show that duplication of *Dscam* is neither necessary or sufficient to cause CHDs, since the gene is not within the minimal *Ifnar1 – Kcnj6* region identified by Liu et al as being sufficient for CHDs (*Liu et al., 2013*) but is duplicated in the Dp6Tyb strain which we show here does not have CHDs. Taking the human and mouse data together, this suggests that several dosage-sensitive genes may contribute to CHDs, with potentially no single gene being absolutely required.

Dp1Tyb mice show AVSD with ventricular but not atrial shunting. Analysis of AVSD in people with DS showed that the most common subtype was a complete AVSD with both atrial and ventricular shunting and the next most common was AVSD with exclusive ventricular shunting similar to that

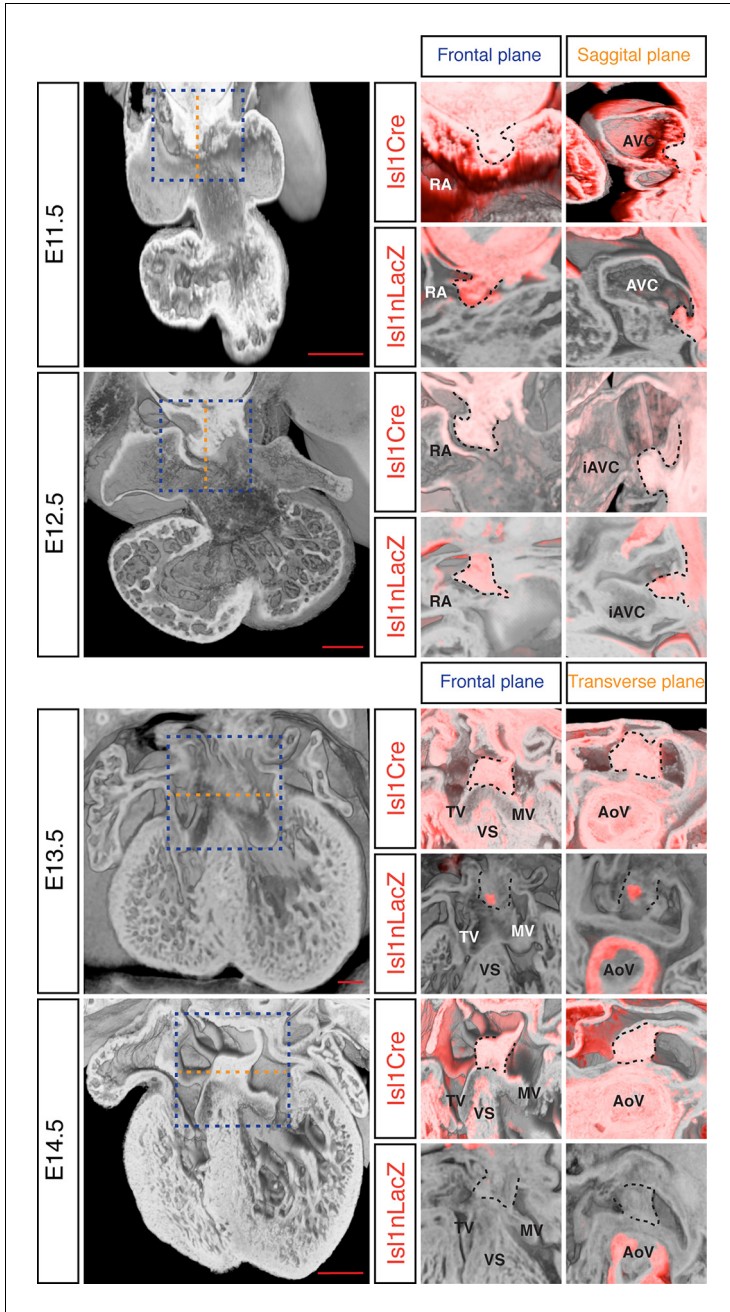

**Figure 4.** Development of the DMP. Left panels show a series of 3D four-chamber views of hearts at embryonic stages E11.5, 12.5, 13.5 and 14.5. Middle panels show a close up of the AV junction (frontal plane view corresponds to blue dashed box in the left panels) in the Isl1Cre/Rosa26RLacZ (Isl1Cre) and the Isl1nLacZ strains; β–galactosidase expression is pseudo-colored in red. Right panels show a sagittal view of the AV junction in E11.5 and E12.5 hearts and a transverse view in E13.5 and E14.5 hearts; sagittal and transverse planes shown as orange dashed line in left panels; β–galactosidase expression is pseudo-colored in red. In the middle and right panels the DMP is marked with a black dashed contour. The number of biological replicates used for each of the developmental stages, E11.5, E12.5, E13.5 and E14.5 respectively, were: n = 2, 2, 6 and 4 for the Isl1Cre/Rosa26RLacZ and n = 10, 9, 7 and 5 for the Isl1nLacZ. AoV, aortic valve; AVC, atrioventricular canal; iAVC, inferior atrioventricular cushion; MV, mitral valve; RA, right atrium; TV, tricuspid valve; VS, ventricular septum. Scale bar, 200 μm.

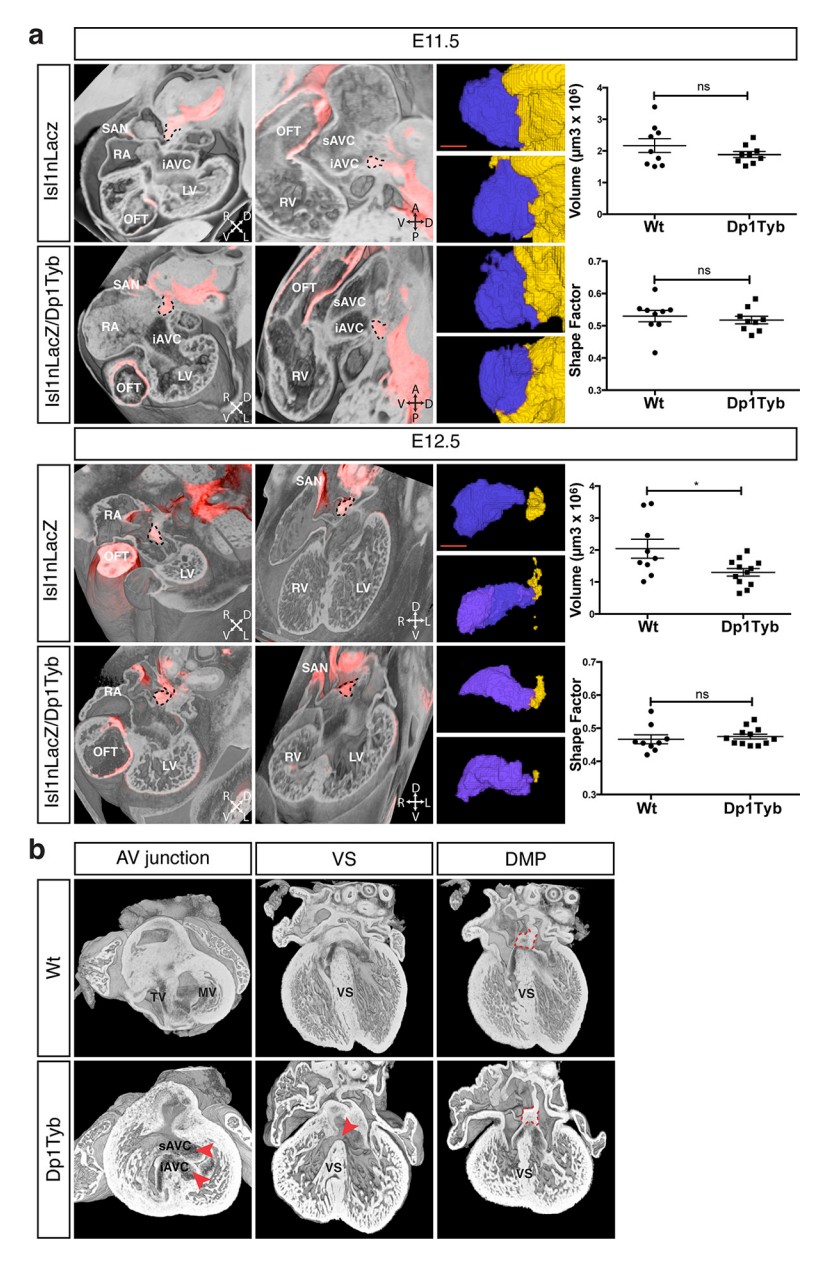

**Figure 5.** Largely normal development of the DMP in Dp1Tyb mice. (a) 3D reconstruction of control Isl1nLacZ and Isl1nLacZ/Dp1Tyb E11.5 and E12.5 hearts analyzed using dual filter HREM. In the first two columns heart morphology is in grey and LacZ expression is pseudo-colored in red. Left panels show a transverse view of the developing heart and middle panels show a sagittal plane at E11.5 and a 4-chamber view at E12.5. The DMP is marked with a dashed black contour. Orientation is marked with 4 arrows (A, anterior; D, dorsal; L, left; P, posterior; R, right; V, ventral). Right image panels show two representative 3D reconstructions of the DMP for control and Dp1Tyb embryos at E11.5 and E12.5; the DMP is pseudo-colored in blue and surrounding dorsal mesenchyme is pseudo-colored in yellow. Graphs show the volume and shape factor of the DMP (n = 9 Wt and 9 Dp1Tyb E11.5 embryos, n = 9 Wt and 12 Dp1Tyb E12.5 embryos). Results are presented with measurements from each heart and as mean ± s.e.m. Statistical analysis with an unpaired Student's t-test; ns, not significant, *P<0.05. (b) 3D HREM rendering of hearts from a single Wt and a single Dp1Tyb embryo at E14.5. The left images show the AV junction in short axis view from the ventricles towards the atria. Red arrowheads indicate the bridging leaflets of the common AV junction in the Dp1Tyb heart. The middle images show a 4-chamber view visualizing the ventricular septum. Red arrowhead indicates a pVSD in the Dp1Tyb heart. The right images show a more dorsal plane of the 4-chamber view to visualize the DMP, marked with a red dashed contour in both Wt and Dp1Tyb hearts. iAVC, inferior atrioventricular cushion; LV, left ventricle; MV, mitral valve; OFT, outflow tract; RA, right atrium; RV, right ventricle; SAN, sinoatrial node; sAVC, superior atrioventricular cushion; TV, tricuspid valve; VS, ventricular septum. Scale bar, 50 µm.

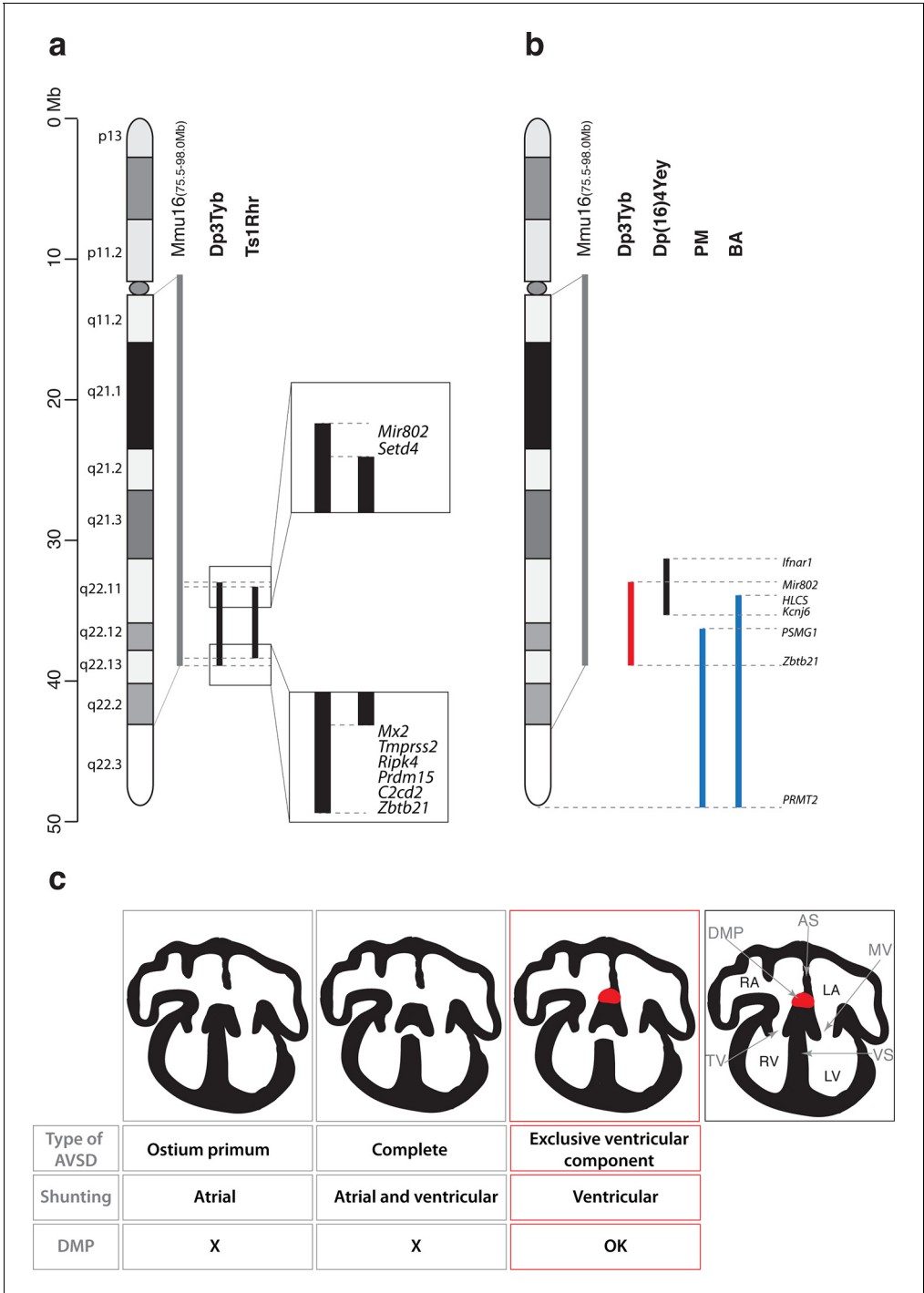

**Figure 6.** Genetic and mechanistic dissection of CHD in mouse models of DS. (a) Representation of Hsa21 and the conserved region of synteny with the telomeric part of Mmu16 (grey line). Black lines indicate the extent of the duplications in Dp3Tyb and Ts1Rhr strains. Magnifications show the ends of the duplications indicating the 8 genes duplicated in Dp3Tyb but not Ts1Rhr mice. (b) Hsa21 and orthologous portion of Mmu16 (grey line), showing extent of duplication in Dp3Tyb (red line, this study) and Dp(16)4Yey (black line) (*Liu et al., 2013*). Also shown are the extent of partial trisomies of two individuals with DS who had a VSD (PM) or AVSD (BA) (blue lines) (*Korbel et al., 2009*). (c) Diagrams show a normal heart (right) and 3 types of AVSD depending on the relationship of the AV valves to the septal components and the presence/absence of the DMP (red). Defects in the DMP are seen in AVSD with atrial shunting (ostium primum) or in complete AVSD with both atrial and ventricular shunts. Dp1Tyb mice have AVSD with a ventricular shunt and the DMP is present (red boxes). AS, atrial septum; DMP, dorsal mesenchymal protrusion; LA, left atrium; LV, left ventricle; MV, mitral valve; RA, right atrium; RV, right ventricle; TV, tricuspid valve; VS, ventricular septum.

seen in the Dp1Tyb mouse (*Freeman et al., 2008*). Thus, although Dp1Tyb mice show AVSD, they only model one subtype of AVSD seen in people with DS – AVSD with ventricular shunting. It is not known why some cases of DS have complete AVSD whereas others have subtypes with only atrial or ventricular shunting, but it is tempting to speculate that there are distinct mechanisms driving defects in the atrial and ventricular septa.

AVSD have been shown to result from defects in development of the DMP (*Briggs et al., 2013*; *Goddeeris et al., 2008*; *Rana et al., 2014*; *Tian et al., 2010*; *Webb et al., 1999*; *Xie et al., 2012*). Importantly, a variety of different genetic manipulations that affect the DMP all resulted in AVSD with exclusive atrial shunting, demonstrating the importance of the DMP for the correct formation of the atrial side of the AV junction (*Briggs et al., 2012*). In contrast, in Dp1Tyb mice we found normal DMP development and no AVSD with atrial shunting, but instead saw exclusively AVSD with ventricular shunting (*Figure 6c*). Thus we propose that defects in the atrial septum causing the ostium primum type of AVSD are caused by malformation of the DMP, whereas ventricular defects are due to perturbations in other tissues, such as endocardial cushions or myocardium. Interestingly, Gata4-deficient mice display AVSD and have a normal DMP, but show perturbations in myocardial signaling to the adjacent atrioventricular cushion mesenchyme, suggesting that defects in myocardial signaling could lead to AVSD (*Misra et al., 2014*; *Rajagopal et al., 2007*). To our knowledge the Dp1Tyb and Dp3Tyb mouse strains are the first to have been shown to have AVSD with ventricular shunting and thus will allow direct studies of the pathological mechanisms underlying this subtype of AVSD.

In conclusion, we have generated a mapping panel of 7 mouse strains that can be used to identify dosage-sensitive genes underlying the broad range of DS phenotypes, and we have exploited this panel to map the location of genes causing CHD in DS.

## Materials and methods

### Mice

We generated ES cells with a duplication from *Lipi* to *Zbtb21* using Cre/loxP-mediated chromosome engineering (*Yu and Bradley, 2001*) following strategies similar to those previously used to make mice with duplications of Hsa21-orthologous regions (*Brault et al., 2015*; *Li et al., 2007*; *Liu et al., 2013*; *Liu et al., 2011*; *Olson et al., 2004*; *Pereira et al., 2009*; *Yu et al., 2010*). MICER vectors (*Adams et al., 2004*) MHPP352i17 (coordinates of homology region 16:74930370–16:74937378 Mb, mouse assembly GRCm38/mm10) and MHPN352i16 (16:97977263–16:97982380 Mb) were used sequentially as targeting vectors to insert loxP sites proximal to *Lipi* and distal to *Zbtb21* respectively in HM-1 ES cells (*Magin et al., 1992*); the cells had been tested to be negative for mycoplasma contamination. Targeting was carried out by standard procedures. Cre recombinase was transiently expressed in double-targeted ES cells to induce recombination between the loxP sites. Segmental duplication was confirmed by Southern blot analysis, targeted clones were injected into blastocysts to generate chimeric mice and these were bred to establish the C57BL/6J.129P2-Dp(16Lipi-Zbtb21)1TybEmcf/Nimr (Dp1Tyb) mouse strain by standard methods. ES cells with a duplication from *Mis18a* to *Runx1* were generated in the same way using MICERs MHPP323h04 (16:90563769 – 16:90577148 Mb) and MHPN219i02 (16:93054020 – 16:93062456 Mb) to target loxP sites proximal to *Mis18a* and distal to *Runx1* respectively, followed by Cre-mediated recombination, and correctly targeted ES cells were used to establish the C57BL/6J.129P2-Dp(16Mis18a-Runx1)2TybEmcf/Nimr (Dp2Tyb) mouse strain. For the remaining duplication strains C57BL/6J.129P2-Dp(16Mir802-Zbtb21)3TybEmcf/Nimr (Dp3Tyb), C57BL/6J.129P2-Dp(16Mir802-Dscr3)4TybEmcf/Nimr (Dp4Tyb), C57BL/6J.129P2-Dp(16Dyrk1a-B3galt5)5TybEmcf/Nimr (Dp5Tyb), C57BL/6J.129P2-Dp(16Igsf5-Zbtb21)6TybEmcf/Nimr (Dp6Tyb), and C57BL/6J.129P2-Dp(16Lipi-Hunk)9TybEmcf/Nimr (Dp9Tyb) we used an in vivo Cre-mediated recombination strategy whereby we bred female mice containing the $Hprt^{tm1(cre)Mnn}$ allele (*Tang et al., 2002*) and two loxP sites located in trans configuration on Mmu16 at the boundaries of the desired duplication, to C57BL/6JNimr males and Cre activity in the female germline from the $Hprt^{tm1(cre)Mnn}$ allele resulted in occasional pups (0.7–6%) with recombination between the loxP sites generating the duplication. The loxP sites were derived from targeting with the 4 MICERs described above as well as MHPP432c09 (16:94538615–16:94546849 Mb) located between *Dscr3* and *Dyrk1a* and MHPN235b18 (16:96327324–16:96331804 Mb) located between *B3Galt5* and *Igsf5*.

Dp(16Cbr1-Fam3b)1Rhr (Ts1Rhr) mice (*Olson et al., 2004*), *Isl1*<sup>tm3Sev</sup> (Isl1nLacZ) (*Sun et al., 2007*), *Isl1*<sup>tm1(cre)Sev</sup> (Isl1Cre) (*Cai et al., 2008*), Gt(ROSA)26Sor<sup>tm1Sor</sup> (ROSA26RLacZ) (*Soriano, 1999*), *Hprt*<sup>tm1(cre)Mnn</sup> and all duplication mouse strains generated above were maintained by backcrossing to C57BL/6JNimr. All mice used for experiments had been backcrossed for at least 5 generations. Specifically, Dp1Tyb was analyzed after backcrossing to C57BL/6JNimr for 6–9 generations (N6-N9), Dp2Tyb at N5, Dp3Tyb at N8, Dp4Tyb at N5-N7, Dp5Tyb at N5, Dp6Tyb at N5-N11, Dp9Tyb at N11 and Ts1Rhr at N11-N12. Genotyping was carried out using custom probes (Transnetyx). All animal work was carried out under a Project Licence granted by the UK Home Office.

## Enumeration of genes

To count numbers of genes in the duplication intervals we used the Biomart feature of Ensembl (mouse genome assembly GRCm38.p4) to count numbers of genes within a given interval, filtering either for protein-coding genes or for protein-coding genes with orthology to human. In addition we included the Mx2 gene as a protein-coding gene, since Ensembl did not automatically classify this gene as coding.

## Array comparative genome hybridization (aCGH)

Genomic DNA was prepared from the tail of each of the duplication strains and from C57BL/6JNimr mice to be used as a reference control using either phenol-chloroform extraction or DNeasy Blood and Tissue Kit (Qiagen, UK). DNA (1 µg) was analyzed by Roche Diagnostics Limited using a mouse $3 \times 720$ K array (Roche NimbleGen) or by Oxford Gene Technology using a mouse $1 \times 1$ M array (Agilent Technologies). 50–75 mers probes were used and the design was based on the genome assembly mm9. The hybridized aCGH slides were scanned for Cy3 (test) and Cy5 (control) channels. The Log2 ratios of the test/control were calculated and plotted as graphs using Prism 7.

## HREM imaging and 3D modeling

E14.5 embryonic hearts were dissected and fixed for 30 min in 4% paraformaldehyde followed by a 1 hr wash in distilled water and secondary fixation overnight. Fixed samples were dehydrated and embedded in modified JB4 methacrylate resin (*Weninger et al., 2006*) and sectioned at 2 µm. HREM imaging (isometric resolution of 2 µm) used a Hamamatsu Orca-HR camera. Data sets were normalized and subsampled prior to 3D volume rendering using Osirix v5.6 (*Rosset et al., 2004*). As the expected CHD were not fully penetrant, a minimum of 17 E14.5 mutant embryos were compared to littermate controls. Phenotype analysis was performed blind for genotype, and classification of type of CHD was carried out as previously described (*Dunlevy et al., 2010*). For dual-wavelength HREM, a conventional Xgal reaction was performed followed by 4% paraformaldehyde fixation, dehydration and embedding, and imaging was carried out using a Jenoptik ProgRes C14 camera with dual filter (59022bs, Chroma Technology Corp). Image analysis of the DMP was done using ITKsnap 2.4.0 and Volocity 6.2.1 software packages. Volocity was used to calculate the volume of the DMP and the shape factor (shape factor is 1 for a perfect sphere and <1 for more irregular shapes). A minimum of 9 biological replicates per group was analyzed and analysis of the DMP was performed blind for genotype.

## Acknowledgements

We thank Yann Herault for reagents and advice, Roger Reeves and Sylvia Evans for mice, Mauro Tolaini and the Procedural Services section of the MRC National Institute for Medical Research for embryo injections, Laurent Dupays for collecting the Isl1Cre/R26LacZ mice, Robert Wilson for image and software assistance and Bob Anderson for anatomical advice. This project was supported by grants from the Wellcome Trust (grant numbers 080174 and 098328), by MRC programme U117527252 and by the Francis Crick Institute which receives its core funding from the UK Medical Research Council, Cancer Research UK and the Wellcome Trust.

# Additional information

## Funding

| Funder | Grant reference number | Author |
|---|---|---|
| Wellcome Trust | 080174 | Elizabeth MC Fisher<br>Victor LJ Tybulewicz |
| Medical Research Council | U117527252 | Victor LJ Tybulewicz |
| Wellcome Trust | 098328 | Elizabeth MC Fisher<br>Victor LJ Tybulewicz |

The funders had no role in study design, data collection and interpretation, or the decision to submit the work for publication.

## Author contributions

Eva Lana-Elola, Conception and design, Acquisition of data, Analysis and interpretation of data, Drafting or revising the article; Sheona Watson-Scales, Conception and design, Acquisition of data, Analysis and interpretation of data; Amy Slender, Dorota Gibbins, Alexandrine Martineau, Charlotte Douglas, Acquisition of data, Analysis and interpretation of data; Timothy Mohun, Conception and design, Analysis and interpretation of data, Drafting or revising the article; Elizabeth MC Fisher, Victor LJ Tybulewicz, Conception and design, Drafting or revising the article

## Author ORCIDs

Victor LJ Tybulewicz [iD] http://orcid.org/0000-0003-2439-0798

## Ethics

Animal experimentation: All animal work was carried out under a Project Licence granted by the UK Home Office.

## Decision letter and Author response

Decision letter https://doi.org/10.7554/eLife.11614.sa1
Author response https://doi.org/10.7554/eLife.11614.sa2

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
