## [Decision Letter]

Thank you for submitting your work entitled "Genetic dissection of Down syndrome-associated congenital heart defects using a new mouse mapping panel" for consideration by *eLife*. Your article has been reviewed by three peer reviewers, and the evaluation has been overseen by a Reviewing Editor and Stylianos Antonarakis as the Senior Editor.

The reviewers have discussed the reviews with one another and the Reviewing editor has drafted this decision to help you prepare a revised submission.

This manuscript describes the cardiac phenotype in seven mouse strains with partial trisomies of regions of mouse chromosome 16. These morphological analyses enable the identification of a minimal critical region which, when trisomic, causes congenital heart defects. It is further reported that there are two loci within this region that are required to be trisomic for the effect on heart development. In addition, the authors use the abnormal phenotype to show that AVSD with shunting at the ventricular level can develop in the presence of a morphologically normal dorsal mesenchymal protrusion (DMP).

The reviewers appreciated the sophisticated genetic manipulations and the high quality of the morphological data. However the following points need to be addressed:

1) In the comparison between human and mouse more discussion is required of the extent to which the studies of Down Syndrome translocation cases and the intervals defined by them match the mouse data presented here. The paper by Korbel et al. in PNAS 2009 is mentioned but the authors could be a bit clearer on the degree of genetic content overlap between the regions they define in their manuscript and the <2-Mb DS-specific congenital heart disease (DSCHD) interval defined by Korbel.

There is a clear difference in phenotype that the authors discuss but rather underplay (“Dp1Tyb mice show AVSD with ventricular but not atrial shunting” – “it is tempting to speculate that there are distinct mechanisms driving defects in the atrial and ventricular septa”). About 80% of Down's hearts with AVSD have the "complete" type, with shunting at atrial and ventricular level, whereas none of the hearts from mouse partial trisomies had this phenotype. The authors speculate that different mechanisms might act in the atrial and ventricular aspects of AV septal formation. If so, it could be considered that their models encompass only one of those two mechanisms.

There is no mention in the Discussion of mouse trisomy 16, which is not a good model of Down's-type defects for other reasons, but Ts16 hearts do have AVSD with shunting at the atrial level (Webb et al., (1999) Circ. Research 84:897-905), which is what is absent in the partial trisomy models studied here. Interestingly, Ts16 hearts do have deficiencies in DMP development.

2) CHD are observed in 50% of people with Down Syndrome and usually necessitate cardiac surgery to avoid any late impairment. Here the results are limited to 14.5dpc embryos in which CHD and mainly AVSD are observed but little is described about the consequence of the CHD on perinatal viability. Kamp et al. (2010; PMID20511334) reported the identification of mouse congenital heart disease loci with most of them inducing perinatal lethality with CHD at birth. If the authors have data on the frequency of perinatal lethality and structural heart defects in the duplication carrier at birth, these should be included.

3) It would be interesting to see a table with the number of individuals studied showing the different types of CHD compared to the total of embryos in the different mouse cohorts from the 7 duplication.

The authors should also discuss the variability of the CHD and AVSD phenotypes observed in wt control animals analysed from the different cohorts of mice. For example, the wt in Dp5Tyb and DP6Tyb or Ts1Rhr showed almost 20% of CHD compared to Dp9Tyb (5%?) Dp2Tyb and Dp3Tyb (10%?). What are the origins of such discrepancies in wt cohorts? A comparison of the CHD in wild-type individuals from the different cohorts, presented as a supplementary table, would add value to the analysis.

4) The authors have done their analysis after 5 backcrosses on the C57BL/6JNimr genetic background. Could they assess if this level is sufficient in this particular situation? The presence of 2 copies of the 129S8 allele on the duplicated fragment versus the B6 allele may influence the outcome of the study (and may also explain the heterogeneity of the control littermates). Thus it would be interesting to check what is the real contribution of the B6 and 129 alleles, at least for the trisomic regions in the different models.

5) An important point is whether by crossing strains Dp4Tyb, Dp5Tyb and Dp6Tyb, the CHD can be reconstituted to narrow further the key genetic contributions. Incomplete penetrance is commonplace in CHD and the finding that two genes require three copies to produce CHD in these models is intriguing in this context and merits further discussion.

6) The conclusions about which regions of Dp1Tyb are required in three copies to induce Downs-type heart defects rely wholly on comparisons between some very small numbers of abnormal hearts, which are usefully summarized in Table 1. Hearts from Ts1Rhr and Dp5Tyb each had 2 AVSD (of 21 and 20 examined respectively), which is undoubtedly statistically different from the incidences in Dp1Tyb and Dp3Tyb, but it is difficult to say with certainty that there is no effect in Ts1Rhr and Dp5Tyb, particularly given that there was only 1 AVSD observed in about 170 wild type hearts. There could be a weak combinatorial effect of a number of genes throughout the region. The authors should be more cautious in the wording of their conclusions. [One of the AVSD in Dp5Tyb did not have an ASD or VSD, an outlier which goes without explanation or mention].

7) For completeness, it would be useful to generate Isl1Cre lineage marked Dp1Tyb hearts, as well as the Isl1nLacZ expression (this is incorrectly termed "lineage" in the subsection “The DMP is present and largely unaffected in the Dp1Tyb mouse model for DS”) marked hearts. It would be informative to show these data at E14.5, in definitively AVSD hearts.

---

## [Author Response]

1) In the comparison between human and mouse more discussion is required of the extent to which the studies of Down Syndrome translocation cases and the intervals defined by them match the mouse data presented here. The paper by Korbel et al.in PNAS 2009 is mentioned but the authors could be a bit clearer on the degree of genetic content overlap between the regions they define in their manuscript and the <2-Mb DS-specific congenital heart disease (DSCHD) interval defined by Korbel.

We have amended the Discussion to include a fuller discussion of the CHD in humans with partial trisomy of Hsa21 as evaluated by Barlow et al. (2001) and Korbel et al. (2009), comparing their conclusions to our data and that of Liu et al. (2013) in the mouse (Discussion, third paragraph). To aid in this we have added a new panel to Figure 6 (Figure 6B) showing the extent of the shortest partial trisomies in humans with DS that lead to VSD or AVSD respectively. The figure also shows the shortest duplication that leads to CHDs in the mouse as reported in the study by Liu et al. (2013).

The 1.77 Mb region reported in Korbel 2009 as being a minimal critical region for CHD was derived from a region present in all individuals with partial trisomy of Hsa21 who had CHDs which was further limited to the overlap with the orthologous region on Mmu16, since mice with a duplication of just this orthologous region have CHDs (Dp1Yey mice, Li et al. 2007). In other words, this 1.77 Mb region has been identified based on several individuals and on mouse data, and there is no known individual trisomic just for this 1.77 Mb region who has CHD. In view of this, we thought it was more useful to show the extent of trisomy in two individuals (PM and BA) reported in Korbel 2009 who had the shortest trisomies leading to VSD and AVSD (7.7 Mb and 10.0 Mb respectively). As can be seen in Figure 6B, these two regions overlap partly with the Dp3Tyb region reported in our study, but the trisomy in individual PM does not overlap with the *Ifnar1-Kcnj6* region identified by Liu et al. 2013. Taken together this suggests that several dosage-sensitive genes may contribute to CHDs, with potentially no single gene being absolutely required.

There is a clear difference in phenotype that the authors discuss but rather underplay (“Dp1Tyb mice show AVSD with ventricular but not atrial shunting” – “it is tempting to speculate that there are distinct mechanisms driving defects in the atrial and ventricular septa”). About 80% of Down's hearts with AVSD have the "complete" type, with shunting at atrial and ventricular level, whereas none of the hearts from mouse partial trisomies had this phenotype. The authors speculate that different mechanisms might act in the atrial and ventricular aspects of AV septal formation. If so, it could be considered that their models encompass only one of those two mechanisms.

We agree with the reviewers that Dp1Tyb models only one subtype of AVSD found in DS (with ventricular but not atrial shunting). As requested, we have added a sentence in the Discussion to clearly emphasize this point (“Dp1Tyb mice show AVSD with ventricular but not atrial shunting. Analysis of AVSD in people with DS showed that the most common subtype was a complete AVSD with both atrial and ventricular shunting and the next most common was AVSD with exclusive ventricular shunting similar to that seen in the Dp1Tyb mouse (Freeman et al., 2008)”).

There is no mention in the Discussion of mouse trisomy 16, which is not a good model of Down's-type defects for other reasons, but Ts16 hearts do have AVSD with shunting at the atrial level (Webb et al., (1999) Circ. Research 84:897-905), which is what is absent in the partial trisomy models studied here. Interestingly, Ts16 hearts do have deficiencies in DMP development.

As noted by the reviewers, Ts16 is not a good model for DS heart defects because it is trisomic for large numbers of genes that are not orthologous to genes on Hsa21. Indeed only 25% of Mmu16 is orthologous to Hsa21. Nonetheless this is an interesting mouse strain because it has defects in DMP development and shows AVSDs. We now refer to the analysis of Ts16 mice and the AVSD and DMP defects (Webb et al. 1999) in the Introduction (second paragraph) and Discussion (fourth paragraph).

2) CHD are observed in 50% of people with Down Syndrome and usually necessitate cardiac surgery to avoid any late impairment. Here the results are limited to 14.5dpc embryos in which CHD and mainly AVSD are observed but little is described about the consequence of the CHD on perinatal viability. Kamp et al. (2010; PMID20511334) reported the identification of mouse congenital heart disease loci with most of them inducing perinatal lethality with CHD at birth. If the authors have data on the frequency of perinatal lethality and structural heart defects in the duplication carrier at birth, these should be included.

We agree this is an interesting point, but we do not have data on structural defects at birth. We estimate it would take at least 4 months to generate this. However, we have data on lethality of the duplication strains. At weaning we recover only 50% of the expected number of Dp1Tyb pups, and 75% of the expected number of Dp3Tyb pups. Given that in both strains around 25% of embryos have the most severe AVSD defects, this is the most likely cause of the lethality in Dp3Tyb, and explains part of the lethality in Dp1Tyb. The additional lethality in Dp1Tyb is due to other causes, including hydrocephalus. We have added data on the recovery of pups from each of the 7 strains in a new Table 1, present the data in the Results (subsection “A mouse genetic mapping panel for DS”) and comment on it in the Discussion (first paragraph).

3) It would be interesting to see a table with the number of individuals studied showing the different types of CHD compared to the total of embryos in the different mouse cohorts from the 7 duplication.

This data is shown in Table 2 (used to be Table 1).

The authors should also discuss the variability of the CHD and AVSD phenotypes observed in wt control animals analysed from the different cohorts of mice. For example, the wt in Dp5Tyb and DP6Tyb or Ts1Rhr showed almost 20% of CHD compared to Dp9Tyb (5%?) Dp2Tyb and Dp3Tyb (10%?). What are the origins of such discrepancies in wt cohorts? A comparison of the CHD in wild-type individuals from the different cohorts, presented as a supplementary table, would add value to the analysis.

Again, this data is shown in Table 2 (used to be Table 1), with the numbers of embryos showing specific types of defects shown against each of the 7 strains, for both WT and mutant embryos.

For convenience, the Table below shows a summary of all defects found in the WT embryos from all 7 strains together.

**Defect****Number of embryos****% embryos**ASD21.2%VSD2012%OFT+VSD31.8%AVSD+VSD21.2%Total defects2716%**Total number of WT embryos**164100%

We can see that the incidence of AVSDs in the WT embryos was very low (1.2%) and the most prevalent defect was a VSD (seen in 15.2%). These VSDs were generally very small and most likely represent slight developmental delay – developmental timing could vary between litters, giving the apparent differences in defects between strains.

4) The authors have done their analysis after 5 backcrosses on the C57BL/6JNimr genetic background. Could they assess if this level is sufficient in this particular situation? The presence of 2 copies of the 129S8 allele on the duplicated fragment versus the B6 allele may influence the outcome of the study (and may also explain the heterogeneity of the control littermates). Thus it would be interesting to check what is the real contribution of the B6 and 129 alleles, at least for the trisomic regions in the different models.

All the analysis was carried out with embryos that had been backcrossed at least for 5 generations into B6, but in some cases much more. Specifically, Dp1Tyb was analyzed after backcrossing to C57BL/6JNimr for 6-9 generations (N6-N9), Dp2Tyb at N5, Dp3Tyb at N8, Dp4Tyb at N5-N7, Dp5Tyb at N5, Dp6Tyb at N5-N11, Dp9Tyb at N11 and Ts1Rhr at N11-N12. Note that the strains showing significant CHDs were analyzed at N6-N9 (Dp1Tyb) and N8 (Dp3Tyb). This information has been added to the Materials and methods (subsection “Mice”, last paragraph).

The reviewers ask if this level of backcrossing is sufficient and if the phenotype can be affected by the presence of 129P2 (not 129S8) alleles at the duplication. These are interesting issues but we are not able to address them directly. The first would require us to repeat the analysis at different backcross generations, which we have not done. We can point to our earlier study with Tc1 mice where we found the frequency of CHDs after N2 backcross to B6 was higher than that found in embryos on a 129xB6 F1 background (Dunlevy 2010), implying that more backcrossing to B6 makes the phenotype stronger. In the studies reported here the backcrossing was much more extensive (N6-N9 for Dp1Tyb, N8 for Dp3Tyb), but we don’t know if the phenotype would get even stronger with further backcrosses.

Regarding the contribution of the 129P2 alleles at the duplication, again we cannot judge this. The strains were all generated on a 129P2 background and then backcrossed. Such backcrossing dilutes away 129P2 alleles everywhere in the genome except around the site of the duplication. The sequences nearest the duplication will always be 129P2-derived, and the extent of remaining 129P2 DNA around this point is variable will depend stochastically on the location of meiotic crossovers. The only way to do a direct comparison would be to generate the duplications in both 129 and B6 ES cells and compare the resulting mice, something we have not done.

Finally, the reviewers ask if the extent of backcrossing could account for the difference in the incidence of defects recorded in WT mice. We don’t believe this can explain the variation in the frequency of defects. For example higher levels of defects were found in the WT mice from the Dp5Tyb (N5) and Ts1Rhr (N11-N12) strains and no defects at all in WT mice in Dp4Tyb (N5-N7) mice. So there appears to be no correlation between extent of backcross and incidence of defects in WT mice. Instead we think the large majority of defects seen in WT mice are very small VSDs, which are caused by slight developmental delay (see above).

5) An important point is whether by crossing strains Dp4Tyb, Dp5Tyb and Dp6Tyb, the CHD can be reconstituted to narrow further the key genetic contributions.

We agree that this is certainly interesting, and we are in the process of doing this, in order to get a better idea of where the key dosage-sensitive genes lie. These data will be used in a follow up paper and we estimate that it will take at least another 12 months before we would have sufficient data on heart structure in crosses of Dp4Tyb x Dp5Tyb, Dp4Tyb x Dp6Tyb and Dp5Tyb x Dp6Tyb (we need to analyse around 100 embryos from each cross). Of course it might turn out that all three intervals are needed, in which case we might need to make a triple Dp4Tyb x Dp5Tyb x Dp6Tyb cross to reconstitute the phenotype, which would take longer still. While interesting, this would give more information on which intervals contain dosage-sensitive genes, but would not identify the genes themselves. Thus in our view it would only be an incremental advance over what is presented in the current manuscript, and does not warrant the very long delay that would be incurred by waiting for this data.

Incomplete penetrance is commonplace in CHD and the finding that two genes require three copies to produce CHD in these models is intriguing in this context and merits further discussion.

We have added text to the Discussion about incomplete penetrance of CHD seen in both humans with DS and in the mouse models (first paragraph). However, we are not sure why the reviewers think a requirement for 2 dosage-sensitive genes to be present in 3 copies would contribute to incomplete penetrance.

6) The conclusions about which regions of Dp1Tyb are required in three copies to induce Downs-type heart defects rely wholly on comparisons between some very small numbers of abnormal hearts, which are usefully summarized in Table 1. Hearts from Ts1Rhr and Dp5Tyb each had 2 AVSD (of 21 and 20 examined respectively), which is undoubtedly statistically different from the incidences in Dp1Tyb and Dp3Tyb, but it is difficult to say with certainty that there is no effect in Ts1Rhr and Dp5Tyb, particularly given that there was only 1 AVSD observed in about 170 wild type hearts. There could be a weak combinatorial effect of a number of genes throughout the region. The authors should be more cautious in the wording of their conclusions.

We agree with the reviewers that it is certainly possible that there is a weak phenotype in Ts1Rhr and Dp5Tyb, which does not reach significance in these studies. As pointed out, each of these two strains showed 2 embryos with AVSD (out of 21 and 20 analyzed respectively). When the incidence of AVSD in these strains is compared to their littermate WT controls, the incidence in the mutants is not significantly different (p = 0.5). We estimate that if these strains have an incidence of AVSD at 10%, we would need to analyze around 60 mutant and 60 wt embryos to see a significance difference at p < 0.05. We have added text to point out the possibility of a weak phenotype in these two strains (Discussion, second paragraph).

[One of the AVSD in Dp5Tyb did not have an ASD or VSD, an outlier which goes without explanation or mention].

We thank the reviewers for pointing this out. We went back to look at the original data from this embryo again and confirmed that it did indeed have an AVSD, but it also had a very small perimembranous VSD, so it is not an outlier. We have reclassified it in Table 2 under “VSD+AVSD”.

7) For completeness, it would be useful to generate Isl1Cre lineage marked Dp1Tyb hearts, as well as the Isl1nLacZ expression (this is incorrectly termed "lineage" in the subsection “The DMP is present and largely unaffected in the Dp1Tyb mouse model for DS”) marked hearts. It would be informative to show these data at E14.5, in definitively AVSD hearts.

We do not have data from such a cross. We estimate that to breed the relevant mice and analyse the embryos could take 9-12 months. However in our view this would not add much to the paper.

The reviewers have asked for this because at E14.5 Isl1 is no longer expressed in the DMP, and so cannot be visualized using LacZ expression from the Isl1nLacZ reporter. The Isl1Cre x R26RLacZ cross referred to by the reviewers is a fate reporter and so LacZ expression persists in the DMP at E14.5. However we do not need to cross this to the Dp1Tyb in order to identify the DMP. We can readily identify the DMP in the 3D HREM data at E14.5 using morphological criteria, in part because we have previously used the Isl1Cre x R26RLacZ fate reporter in wild-type embryos to locate this tissue. Crossing it to Dp1Tyb, would make the study more complete, but, in our view, would not add any further insights into the pathology.

We have corrected the incorrect reference to lineage (subsection “Development of the DMP”).